# Intense Pulsed Light Therapy for Dry Eye Disease: Analyzing Temporal Changes in Tear Film Stability and Ocular Surface between IPL Sessions

**DOI:** 10.3390/healthcare12111119

**Published:** 2024-05-30

**Authors:** Cristina-Patricia Pac, José-María Sánchez-González, Carlos Rocha-de-Lossada, Nadina Mercea, Francis Ferrari, Maria Alexandra Preda, Cosmin Rosca, Mihnea Munteanu

**Affiliations:** 1Department of Ophthalmology, University of Medicine and Pharmacy “Victor Babes”, 300041 Timisoara, Romania; cristina.babut@umft.ro (C.-P.P.); preda.alexandra@umft.ro (M.A.P.); mihnea.munteanu@umft.ro (M.M.); 2Department of Physics of Condensed Matter, Optics Area, University of Seville, 41012 Seville, Spain; 3Department of Ophthalmology, Qvision VITHAS Almeria Hospital, 04120 Almeria, Spain; carlosrochadelossada5@gmail.com; 4Department of Ophthalmology, VITHAS Malaga, 29016 Malaga, Spain; 5Department of Ophthalmology, Regional University Hospital of Malaga, 29009 Malaga, Spain; 6Department of Surgery, University of Seville, Ophthalmology Area, 41009 Seville, Spain; 7Department of Ophthalmology, Municipal Emergency Clinical Hospital, 300254 Timisoara, Romania; nadinamercea@gmail.com; 8Clinique Espace Nouvelle Vision, 6 Rue de la Grande Chaumière, 75006 Paris, France; francis.ferrari@neoris-eyes.com; 9Oculens Clinic, Calea Turzii no. 134-136, 400347 Cluj Napoca, Romania; roscacosmin@yahoo.com

**Keywords:** dry eye disease, intense pulsed light, meibomian gland dysfunction, ocular surface health, evaporative dry eye, tear film stability

## Abstract

Background: Dry eye disease (DED), a prevalent condition with a multifactorial etiology, significantly impacts global health by causing discomfort and visual disturbance. This historical cohort study evaluates the efficacy of Intense Pulsed Light (IPL) therapy on meibomian gland dysfunction (MGD)-related evaporative DED. Methods: The study involved 110 patients (220 eyes) who underwent IPL therapy. Ethical approval was secured, and informed consent was obtained from all participants. A Tearcheck^®^ (ESWvision, Houdan, France) device was used for ocular surface evaluation, measuring tear film stability (NIFBUT, NIABUT), tear film quantity (CTMH, TTMH), and inflammation (OSIE). The study assessed tear film and ocular surface health across multiple IPL sessions. Results: Significant improvements were observed in subjective symptoms (EFT score increased from 29.10 ± 8.87 to 35.91 ± 7.03, *p* < 0.01), tear film stability (NIFBUT increased from 9.37 ± 6.04 to 10.78 ± 5.83 s, *p* < 0.01; NIABUT increased from 11.07 ± 4.98 to 12.34 ± 4.66 s, *p* < 0.01), and tear film surface evaluation (TFSE score decreased from 337.78 ± 414.08 to 206.02 ± 240.44, *p* < 0.01). Tear film quantity remained unchanged (CTMH and TTMH, *p* > 0.05). Conclusions: IPL therapy is a promising treatment for DED, improving symptoms and ocular surface health. Further research is warranted to explore long-term efficacy and optimization.

## 1. Introduction

Dry eye disease (DED) stands as a globally prevalent condition, characterized by a complex interplay of factors that compromise tears and the ocular surface, leading to discomfort, visual disturbance, and potentially significant ocular morbidity [1]. This multifactorial disease, which affects a substantial portion of the global population, represents a significant clinical challenge due to its varied etiology, ranging from age-related changes to environmental influences and post-surgical outcomes [2]. As the search for effective treatment modalities continues, the advent of Intense Pulsed Light (IPL) therapy emerges as a promising approach, especially for those cases primarily driven by evaporative loss related to meibomian gland dysfunction (MGD) [3,4].

The prevalence of DED is underscored by its association with both intrinsic factors, such as aging and hormonal changes, and extrinsic factors like environmental stressors, thereby highlighting the diverse nature of its pathophysiology [5]. Notably, the incidence of DED increases with age, affecting a significant proportion of the elderly population [6]. This age-related increase is attributed to the cumulative effects of environmental exposure, systemic health conditions, and a natural decline in tear production and meibomian gland function over time [7]. Furthermore, surgical interventions on the eye, such as refractive and cataract surgeries, have been identified as potential triggers for DED [8]. These procedures, while aiming to improve visual acuity, can inadvertently disrupt the ocular surface ecosystem, leading to or exacerbating existing DED. The presence of preoperative DED is a known risk factor for exacerbated postoperative symptoms, thus emphasizing the need for careful pre-surgical evaluation and management of ocular surface health to ensure optimal surgical outcomes [9,10].

Evaporative DED, predominantly stemming from MGD, represents the most common subtype of this condition [11]. MGD is characterized by obstruction or dysfunction of the meibomian glands, which are crucial for maintaining the tear film’s lipid layer and preventing excessive tear evaporation [4,12]. Traditional management strategies for MGD and evaporative DED have focused on alleviating obstruction and inflammation through a combination of warm compresses, eyelid hygiene measures, and pharmacological interventions. However, these conventional therapies often fall short in providing long-term relief, prompting the exploration of novel treatment avenues [3].

IPL therapy, originally devised for dermatological applications, has shown remarkable efficacy in managing skin conditions such as rosacea, a chronic inflammatory skin disease that frequently coexists with ocular manifestations, including MGD [13,14,15,16,17]. An unexpected observation that patients receiving IPL for facial rosacea experienced improvements in dry eye symptoms announced a new era in DED management. This discovery has since been substantiated by a growing body of research, affirming IPL’s potential in reducing both the signs and symptoms of DED, particularly in the context of MGD [18,19,20].

The principle behind IPL therapy lies in its ability to deliver broad-spectrum light pulses to targeted areas, inducing selective photothermal damage to aberrant vessels and inflammatory structures without harming surrounding tissues [21]. This mechanism not only addresses the underlying vascular and inflammatory components of MGD but also promotes normalization of meibomian gland function, thereby restoring the ocular surface environment. The application of IPL in DED treatment involves a series of treatments, each consisting of light pulses delivered across the periorbital region, including areas proximal to the meibomian glands [22]. The cumulative effect of these treatments has been documented to yield significant improvements in tear film stability, meibomian gland function, and overall ocular surface health [23].

Despite the encouraging outcomes associated with IPL therapy, the intricacies of its mechanism of action on the ocular surface and meibomian glands warrant further elucidation. The hypothesized benefits include reduction of lid margin telangiectasia, which decreases the release of inflammatory mediators, thermal modulation of meibum consistency, and direct antimicrobial effects, among others [17,24]. These proposed mechanisms align with observed clinical improvements, including enhanced tear film stability and reduced symptoms of eye discomfort and visual disturbance [25,26].

Delving deeper into the application of IPL therapy in the management of DED, it is imperative to consider safety, efficacy, and patient selection criteria to maximize therapeutic outcomes [27,28]. The evolving landscape of DED treatment now incorporates IPL as a significant modality, particularly for patients with MGD-related evaporative DED. The integration of IPL into the treatment repertory for DED signifies a step forward in addressing a condition that imposes a considerable burden on individuals’ quality of life and ocular health. Future research and clinical trials are essential to refine IPL treatment protocols, understand its long-term effects, and establish its position within comprehensive management strategies for DED.

The purpose of this study was to systematically investigate temporal changes in tear film stability and ocular surface health in patients with DED undergoing IPL therapy over multiple treatment sessions, with the aim of elucidating the efficacy of IPL therapy in improving the symptoms and underlying causes of DED between treatment intervals.

## 2. Materials and Methods

### 2.1. Study Design

This research was structured as a historical cohort study, characterized by its retrospective, non-randomized, and singular clinical setting framework. Conducted at the Professor Munteanu Mihnea Eye Clinic in Timisoara, Romania, the study spanned from May 2021 to May 2023. Its design was explicitly chosen to evaluate the intersessional efficacy of IPL therapy on alleviating the clinical manifestations associated with MGD. Importance was placed on the acute response of the ocular surface to the therapy over three specifically timed sessions, offering a detailed perspective on the immediate therapeutic impacts rather than long-term outcomes; a distinct approach from preceding studies. Ocular surface assessments were strategically aligned with the timing of IPL therapy sessions, conducted on Day 1, Day 15, Day 45, and Day 75 with an optional session on Day 105 for cases presenting with more severe conditions, to capture dynamic changes in ocular surface health and provide a thorough understanding of the therapy’s impact.

### 2.2. Ethical Considerations

The research protocol received ethical approval from the Ethics Committee of the Victor Babes University of Medicine and Pharmacy Timisoara, documented under record number 48/2021. This study was conducted in adherence to the principles laid out in the Declaration of Helsinki, ensuring a foundation of ethical integrity and respect for participant rights. Informed consent was obtained digitally from all participants, providing a clear and transparent explanation of the study’s purpose, the procedures involved, and the use of clinical data for scholarly dissemination, thereby upholding the highest standards of ethical research practice.

### 2.3. Inclusion and Exclusion Criteria

In the study, a total of 110 patients, encompassing 220 eyes were evaluated. The cohort comprised adult individuals diagnosed with symptomatic MGD who underwent IPL therapy at the clinic during the specified study period. “Symptomatic MGD” refers to patients who were detected with MGD and also reported symptoms associated with DED, including burning, dryness, and similar discomforts.

Inclusion criteria were meticulously aligned with recommendations from the International Workshop on MGD [29], ensuring a targeted and relevant participant selection. Participants were diagnosed with MGD based on the TFOS DEWS II criteria [1,4,30], including symptom screening with questionnaires, and diagnostic tests such as non-invasive break-up time and ocular surface staining.

Exclusion criteria were strictly applied to exclude individuals presenting with any dermatological contraindications to IPL therapy, as well as those who had undergone any alterations in their systemic or ocular management of MGD within six months before or at any time during the study duration, to maintain the purity and reliability of the findings. To ensure the integrity of the study and minimize confounding factors, exclusion criteria were rigorously enforced, following the TFOS Lifestyle Report guidelines [31,32,33]. Participants with systemic pathologies known to affect tear film stability, such as thyroid disease and Sjögren’s syndrome, were excluded. Additionally, individuals who had undergone ocular surgery within the last three months or had experienced other ocular inflammatory conditions, such as uveitis, keratitis, and episcleritis, within the last six months were excluded. Patients with glaucoma, significant skin pathologies—including pigmentation issues, trauma, or cancer— or those who wore contact lenses [34] were not eligible to participate. Individuals taking medications that could influence tear film stability were also excluded [35]. Furthermore, participants who had modified their MGD treatment regimen within six months prior to the study or at any point during its course were excluded to better isolate the effects of IPL therapy on DED associated with MGD.

### 2.4. Procedure Overview

#### 2.4.1. Initial Consultation and Assessment

Before initiating the IPL therapy sessions, participants underwent a comprehensive evaluation to ascertain their current state of vision comfort and to review any potential changes in ocular health. A critical step in the preparatory phase involved assessing the participant’s skin phototype, categorized from I (very fair) to V (dark) [36], to tailor the IPL treatment parameters accurately. Additionally, participants completed a brief questionnaire designed to capture essential information and screen for any contraindications to the forthcoming IPL applications. The IPL treatment was administered using a wavelength range of 500–1200 nm and a fluence of 13 J/cm^2^.

#### 2.4.2. Treatment Application Process

During the treatment sessions, participants were positioned comfortably, with all facial makeup removed to prevent any interference with the therapy. Protective goggles were provided to shield the eyes, followed by the application of a conductive gel on the cheekbone and temporal areas of the face (Figure 1A). The treatment involved administering a sequence of five precise flashes under the lower eyelids, moving from the inner to the outer corner of each eye, ensuring a uniform application of the IPL therapy (Figure 1B). As shown, the patient’s skin condition immediately after the removal of the conductive gel demonstrates the immediate post-treatment state (Figure 1C).

#### 2.4.3. Post-Treatment Care and Observation

IPL therapy, facilitated through the use of Tearstim^®^ technology developed by ESWvision, Houdan, France, was characterized by its non-invasive nature, simplicity, and high safety profile (Figure 2). The post-treatment protocol allowed for the immediate reapplication of makeup, underscoring the non-disruptive nature of the therapy to daily routines. During treatment, skin lesions and moles were covered with special patches. After each IPL treatment session, several safety measures were applied to ensure participant safety and comfort. Participants were instructed to wear sunglasses to protect their eyes from bright light and UV exposure. They were advised to avoid exposing their skin to strong sunlight, sunbeds, or self-tan for at least 2 weeks after treatment. Participants were also instructed to avoid excess heat, such as long baths, spas, steam rooms, and saunas, for at least 24 h or longer if the skin was still red or recovering. Additionally, they were advised to avoid activities involving chlorine, such as swimming, for 48 h post-treatment. Detailed post-treatment care instructions were provided to ensure proper skin care and protection.

#### 2.4.4. Ocular Surface Evaluation Techniques

A foundation of the methodological approach was utilization of the Tearcheck^®^ device (ESWvision, Houdan, France) for a comprehensive evaluation of the ocular surface. This involved a comprehensive assessment using an Eye Fitness Test (EFT) (Appendix A), which measures symptom severity. It is important to note that in the EFT, a higher score indicates a better outcome. The ocular surface evaluation also included measurements of the Central Tear Meniscus Height (CTMH) and Thinnest Tear Meniscus Height (TTMH) for quantifying tear film volume. CTMH and TTMH measurements were taken using the TearCheck^®^ (ESWvision, Houdan, France) device. The procedure involved measuring the tear meniscus height by first instilling fluorescein dye into the eye to highlight the tear film. The device then captured high-resolution images, and used automated algorithms to detect and measure the height of the tear meniscus from these images.

Additionally, a Tear Film Stability Evaluation (TFSE) and an Ocular Surface Inflammatory Evaluation (OSIE) were employed to gauge the level of inflammation and its response to the IPL therapy. The TFSE assesses micro-deformations on the tear film surface, which reflect tear film instability. These deformations are presented in terms of number and intensity during a 10 s imaging period. The tear film of a healthy eye shows very few, low-intensity movements, whereas a patient with DED, linked to a deficiency in the lipid tear film component, shows higher micro-deformations. The frequency and intensity of these deformations are observed throughout the imaging period, allowing for the classification of patients into four categories assigned with score points:

Category 1: a healthy patient with very few, low-intensity micro-deformations.

Category 2: a significant number of micro-deformations grouped towards the end of the 10 s acquisition, regardless of intensity.

Category 3: early onset of micro-deformations with minimal evolution over the 10 s period.

Category 4: early onset of micro-deformations with increasing number and intensity over the 10es period.

The higher the category, the greater the lipid deficiency, with Category 4 patients experiencing the most significant discomfort and unfavorable progression without treatment. Compared to the Non-Invasive Break-Up Time (NIBUT), the TFSE provides a more detailed evolution of tear film behavior over time, showing finer nuances of the tear film surface. The device converts the grade into a score ranging from 18 to 1200 points, providing a detailed and quantifiable assessment of the tear film.

The OSIE utilizes fluorescein dye, which adheres to areas of the ocular surface with alterations due to inflammation. The evaluation is conducted 120 s after instilling fluorescein, allowing for its natural elimination through the tear ducts. In a healthy patient, fluorescein disappears from the ocular surface, showing 0% residual fluorescence. In contrast, in patients with DED, the dye remains in the affected areas beyond 120 s, indicating inflammation. The accuracy of this examination relies on the practitioner’s selections and use of adjustment sliders to evaluate these inflammatory zones accurately.

Also, the Non-Invasive First Break-Up Time (NIFBUT) and Non-Invasive Average Break-Up Time (NIABUT) were measured for assessing tear film stability with a SCHWIND SIRIUS device for corneal pachymetry and topography (SCHWIND eye-tech-solutions GmbH, Kleinostheim, Germany) [37].

The NIFBUT was measured as follows:The NIFBUT is the time interval between the last complete blink and the first appearance of a dry spot or discontinuity in the tear film.The patient was asked to blink naturally, then to keep their eyes open for as long as possible while the device recorded the tear film.The SCHWIND SIRIUS device projected a series of concentric rings onto the cornea and captured high-resolution images to detect the first break in the tear film. The time at which the first break occurred was recorded as the NIFBUT.

The NIABUT was measured as follows:The NIABUT measures the average time taken for multiple tear film break-ups to occur across the corneal surface.Following the same initial procedure, the device continuously monitored the tear film over a specified period, capturing the times at which multiple breaks in the tear film appeared.The average time of these break-ups was calculated and recorded as the NIABUT.

Acceptable Measurement Values:

For healthy individuals, NIFBUT values were typically above 10 s, indicating a stable tear film. In healthy eyes, NIABUT values were generally above 15 s. Lower values for NIFBUT and NIABUT indicate reduced tear film stability, which is often observed in patients with dry eye disease. The device had a cut-off superior point of 17 s.

The ocular surface measured times were Time 1, at the baseline moment, Time 2: during the intermediate IPL session, and Time 3: at the last IPL treatment session.

### 2.5. Statistical Analysis

Data analysis was conducted using SPSS Statistics software, version 29.0, developed by IBM Corporation in Armonk, NY, USA. The size of the study sample was determined with the use of a GRANMO calculator, version 7.12, provided by the Municipal Institute of Medical Research in Barcelona, Spain. This determination was made considering expected two-paired means (repeated in one group), and accepting an alpha risk of 0.05 and a beta risk of 0.2 in the two-sided test. To recognize a statistically significant difference greater than or equal to 0.05, 140 eyes were necessary. The standard deviation was assumed to be 3.8 score points (based on Benitez-del-Castillo et al. [38]) with an anticipated a drop-out rate of 10%. Mean values and standard deviations (SD) were used to summarize continuous data, along with ranges, whereas frequencies (n) and percentages (%) were used for ordinal categorical data.

The analysis involved checking for normal distribution and equal variances before applying either a Student’s *t*-test (for parametric data) or a Wilcoxon signed-rank test (for nonparametric data) for within-group comparisons of clinical outcomes. Comparisons between the groups were made using either an unpaired Student’s *t*-test (parametric) or Mann–Whitney U test (nonparametric). Correlation analyses between variables were conducted using either a Pearson’s correlation coefficient (parametric) or Spearman’s Rho (nonparametric). A stepwise multiple linear regression analysis was utilized to identify factors significantly affecting dry eye symptom changes in the EFT. A significance level of *p* < 0.05 was set for all comparisons [39].

## 3. Results

The participants presented an average age of 51.88 ± 15.26 years, ranging from 18 to 86 years. The demographic distribution of gender among the participants was female, with 69 females (62.7%) and 41 males (37.3%) participating in the study.

The subjective symptom changes (measured with an EFT), tear film stability variations (measured by NIFBUT and NIABUT, and with a TFSE), tear film quantity differences (measured by CTMH and TTMH), and surface evaluation changes (measured by OSIE Type 1 percentage and capture time) are presented in Table 1, which presents the differences between IPL sessions.

In the comprehensive analysis presented in Figure 3, the figure integrates eight box and whisker plots, each illustrating significant findings: (A) EFT scores, (B) NIFBUT, (C) NIABUT, (D) TFSE scores, and Figure 4 (A) Central Tear Meniscus Height (CTMH), (B) Thinnest Tear Meniscus Height (TTMH), (C) OSIE with fluorescein Thilorbin, and (D) OSIE capture time.

In a linear correlation analysis, it was observed that a correlation was detected that was not clinically relevant. A stepwise multiple linear regression analysis was employed to identify factors that significantly influenced changes in dry eye symptoms, as measured by the EFT. The analysis, based on baseline data, yielded an R2 value of 0.09, indicating that 9% of the variance in EFT scores can be explained by the model. The Durbin–Watson statistic was calculated to be 1.72, suggesting a moderate degree of autocorrelation in the residuals of the regression model.

The distribution of additional sessions beyond the initial treatment revealed that a majority of the patients, 77 (70%), did not require any additional sessions. However, 23 patients (20.9%) underwent one additional session, 9 patients (8.2%) required two additional sessions, and only 1 patient (0.9%) needed as many as four additional sessions.

When considering the necessity for a fifth session, it was observed that the majority of the patients, 96 (87.3%), did not require this additional treatment, indicating a positive response to the initial treatment sessions. Conversely, 14 patients (12.7%) were identified as needing a fifth session to achieve the desired therapeutic outcomes.

## 4. Discussion

The results shown in the study indicate a progressive improvement in the management of DED following IPL therapy sessions. Specifically, a notable enhancement was observed in the subjective symptoms of DED, as assessed by improvements in Eye Fitness Test scores from the initial session to subsequent ones. This progression suggests that patients experienced a tangible relief in their symptoms as a direct consequence of the IPL treatments. Further analysis of tear film stability, through measures such as the NIFBUT and NIABUT, revealed a gradual improvement. These findings underscore the therapy’s effectiveness in enhancing the tear film’s stability, which is crucial for the overall comfort and ocular health of individuals suffering from DED. Additionally, the study evaluated the tear film surface and found significant improvements over the course of the IPL sessions. This indicates a positive impact on the quality of the tear film surface, contributing to the alleviation of dry eye symptoms. Despite these positive changes in tear film stability and surface quality, the quantity of the tear film, measured by the CTMH and TTMH, remained unchanged across the sessions. This suggests that while IPL therapy effectively improves the quality of tear film, it does not affect its quantity. The inflammation process in DED was assessed using the OSIE. This method employs fluorescein dye to detect and quantify ocular surface staining, providing a detailed visualization of epithelial damage and potential inflammation. Analysis of fluorescein staining patterns enabled us to evaluate the severity of ocular surface damage, thereby enhancing understanding of the inflammatory state in DED and guiding individualized treatment strategies. The surface evaluation, specifically the OSIE, showed a decrease in the percentage of Type 1 inflammation, marking a significant reduction in ocular surface inflammation from the start to the end of the therapy sessions. However, the consistency in OSIE capture time across the sessions indicates a uniformity in the evaluation process.

The exploration of IPL therapy’s efficacy and safety for treating DED due to MGD has garnered significant interest in ophthalmic research. The study contributes valuable insights into this area, offering empirical evidence that both supports and extends the findings of seminal research in the field. Toyos et al.’s [13] pioneering work in utilizing IPL therapy for DED associated with MGD highlighted the treatment’s potential to significantly improve clinical outcomes, including tear break-up time and patient satisfaction. The results resonate with these findings, showcasing a marked improvement in subjective symptoms and tear film stability across the treatment sessions. This progression underscores the therapy’s potential to offer a tangible benefit to patients suffering from DED due to MGD, reinforcing the value of IPL as a viable treatment option. The safety profile of IPL therapy, a critical component of its clinical application, has been a focus of several studies. Gupta et al. [17] emphasized IPL’s safety, noting an absence of serious adverse events in their cohort. The observations align with this perspective; only minimal adverse events were noted in the study population. These were predominantly mild and transient in nature, such as slight discomfort and redness, further supporting the proposition that IPL therapy offers a favorable safety profile for patients.

The investigation into tear film stability and quality presents findings that echo those of Craig et al. [14] who reported significant improvements in lipid layer grade and noninvasive tear break-up time following IPL treatment. This enhancement in tear film quality is particularly noteworthy, as it directly contributes to alleviating the symptoms of DED. However, the study, similar to the results presented by Albietz et al. [20], observed no significant changes in tear film quantity, suggesting that IPL’s therapeutic effects are more pronounced in improving tear quality rather than volume. This distinction is crucial for tailoring patient expectations and managing treatment outcomes effectively. Furthermore, the reduction in ocular surface inflammation noted in the results aligns with outcomes described by Dell et al. [18,19] who discussed IPL therapy’s efficacy in mitigating signs and symptoms of DED. This concurrence emphasizes the therapy’s role in addressing not only symptomatic relief but also underlying inflammatory processes associated with DED due to MGD.

The findings also parallel the clinical improvements and patient-reported symptom relief detailed by Vora et al. [15] and Vegunta et al. [16], reinforcing the narrative that IPL therapy is an effective intervention for managing DED. These corroborative insights highlight the significance of IPL therapy in the broader context of DED treatment strategies, underscoring its potential to improve patient outcomes significantly.

The minimal side effects reported in the study, which are in line with the safety profiles detailed by Arita et al. [22] and Rong et al. [40], underscore the procedure’s utility in a clinical setting. The predominance of mild adverse events further attests to IPL therapy’s applicability as a safe treatment modality, offering a compelling risk–benefit ratio for patients grappling with DED due to MGD. As the body of evidence grows, as noted by Gianncare et al. [41] and suggested by Vigo et al. [23], further clinical investigations are anticipated to refine patient selection criteria and optimize treatment protocols, thereby enhancing therapeutic efficacy and ensuring the safety of IPL therapy in the management of DED.

### 4.1. Limitations

The study, while contributing valuable insights into the efficacy and safety of IPL therapy for DED due to MGD, is not without limitations. One primary constraint is the retrospective nature of the analysis, which may introduce biases related to patient selection and data collection. Furthermore, the absence of a control group limits the ability to draw causal inferences regarding the observed improvements. Variability in treatment protocols and patient adherence also pose challenges in standardizing the intervention, potentially affecting the generalizability of the findings. A limitation of the study is the absence of measurements for white blood cells, cellular debris, osmolarity, or metalloprotease levels. These parameters could provide a more comprehensive understanding of inflammation in the tear film. Additionally, the reliance on subjective symptom assessments could introduce response bias, highlighting the need for incorporating more objective measures in future studies. The study primarily focused on tear film variables, such as tear film stability and break-up time, to indirectly assess the improvement in MGD following IPL treatment. While increased tear film stability and longer break-up times suggest an improvement in MGD, direct analysis of meibomian gland secretion and expressibility was not conducted. This omission is a limitation of the study.

One limitation of this study is the use of the EFT as the primary assessment tool for symptom severity, instead of more widely validated tests. The EFT was selected because it is the default assessment tool integrated within the Tearcheck^®^ device, ensuring consistency in data collection. While the questions in the EFT mirror those in the ocular surface disease index (OSDI), the scoring system is inverse. Despite this, the comparability of the results remains unaffected due to the identical nature of the questions. We acknowledge that this choice may affect the generalizability of our findings, and future studies should consider using widely validated symptomatology tests to align with common clinical practices and enhance comparability across studies.

### 4.2. Future Lines of Research

Building on the foundational work of Toyos et al. [13], Craig et al. [14], and others [15,16,17,20,23,40,41,42], future research should aim to address the limitations identified in current studies. Investigating the optimal treatment protocol, including the frequency of sessions and the specific parameters of IPL application, would provide valuable insights into maximizing therapeutic outcomes. Furthermore, exploring the underlying mechanisms of IPL therapy’s impact on DED and MGD could enhance understanding of its therapeutic potential. In future studies, the aim is to include measurements of white blood cells, cellular debris, osmolarity, and metalloprotease levels. This will allow us to assess the inflammatory effects of IPL on the tear film more thoroughly and enhance understanding of its therapeutic potential in managing DED. Additionally, research into patient selection criteria would be invaluable in identifying those who stand to benefit the most from this treatment modality.

### 4.3. Practical Application

Despite limitations, the practical applications of IPL therapy in the management of DED due to MGD are promising. The findings, alongside those of Gupta et al. [17] and Albietz et al. [20], suggest that IPL therapy is a safe and effective treatment option that can be incorporated into the current therapeutic arsenal for DED. The procedure’s ability to improve tear film stability and quality, reduce ocular surface inflammation, and enhance patient satisfaction makes it a valuable addition to treatment strategies, especially for patients who have not responded adequately to conventional therapies. In clinical practice, it is essential to consider individual patient characteristics and preferences when recommending IPL therapy, ensuring a personalized approach to DED management. As the evidence base grows and treatment protocols become more refined, IPL therapy is floated to play an increasingly significant role in improving quality of life for patients with DED due to MGD.

## 5. Conclusions

In summary, the results shown from this study highlight the beneficial effects of IPL therapy in improving subjective symptoms, tear film stability, and surface quality for individuals with dry eye disease. The therapy’s impact on reducing ocular surface inflammation further corroborates its potential as a valuable treatment option. Nonetheless, the unchanged measures of tear film quantity suggest a targeted effect of IPL therapy on specific aspects of evaporative dry eye disease, underlining the complexity of its management.

## Figures and Tables

**Figure 1 healthcare-12-01119-f001:**
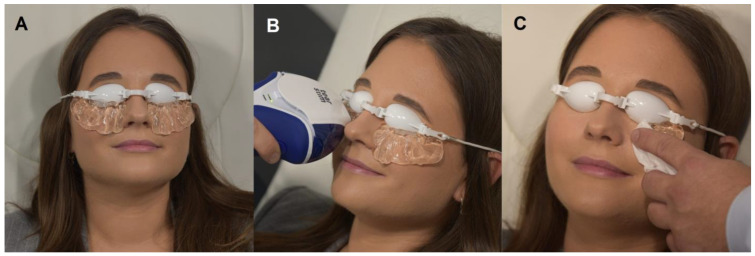
Stages of IPL treatment process: (**A**) patient just prior to treatment, wearing protective eyewear shields with conductive gel applied. (**B**) Application of IPL treatment. (**C**) Patient’s skin condition immediately after removal of conductive gel.

**Figure 2 healthcare-12-01119-f002:**
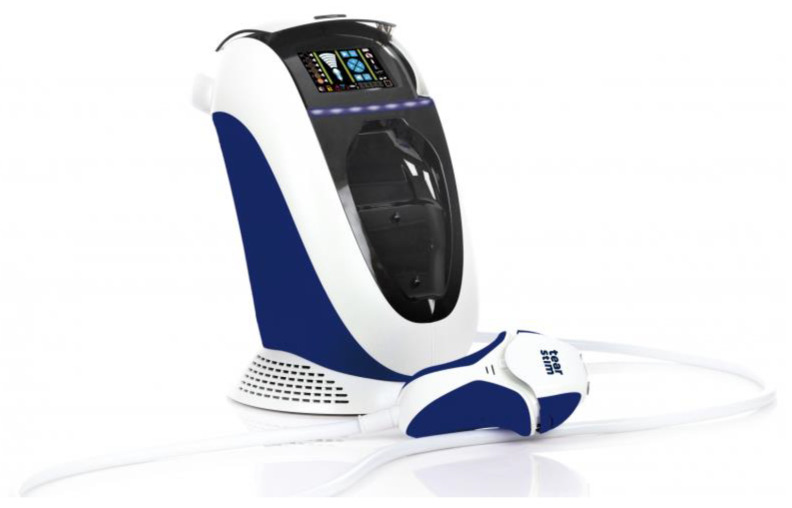
The IPL machine used in the study; Tearstim^®^ (ESWvision, Houdan, France) technology developed by ESWvision, Houdan, France.

**Figure 3 healthcare-12-01119-f003:**
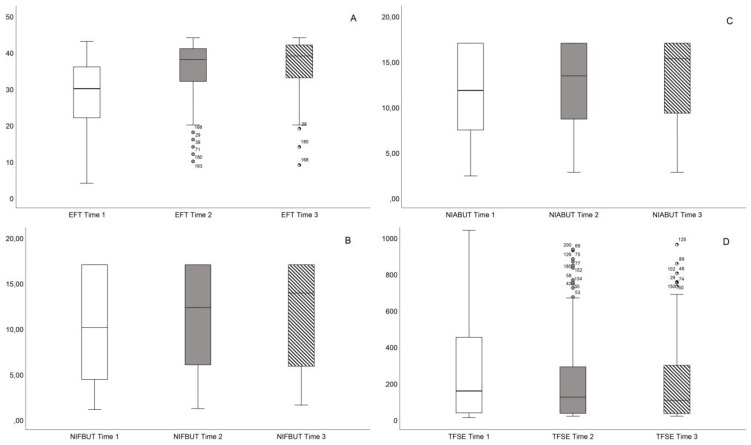
Sequential assessment of ocular health parameters across IPL therapy sessions. (**A**) Eye Fitness Test (EFT) comparison in score points. (**B**) Non-Invasive First Break-Up Time (NIFBUT) analysis in seconds. (**C**) Non-Invasive Average Break-Up Time (NIABUT) distribution in seconds. (**D**) Tear Film Stability Evaluation (TFSE) scores in score points.

**Figure 4 healthcare-12-01119-f004:**
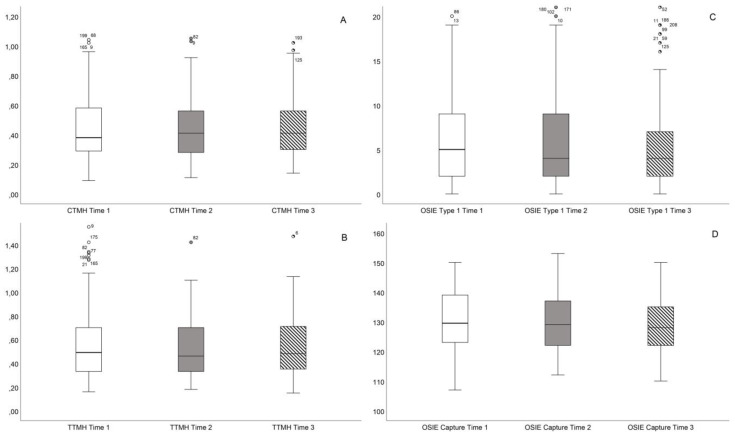
Analysis of tear volume and inflammation through IPL sessions. (**A**) Central Tear Meniscus Height (CTMH) in millimeters. (**B**) Thinnest Tear Meniscus Height (TTMH) in millimeters. (**C**) Ocular surface inflammatory evaluation (OSIE) with fluorescein Thilorbin in percentages. (**D**) OSIE capture time in seconds.

**Table 1 healthcare-12-01119-t001:** Dry eye disease changes between IPL sessions.

Variables	Time 1	Time 2	Time 3	*p* Value ^a^
Subjective Symptoms				
EFT, score points, mean ± SD [range]	29.10 ± 8.87 [4 to 43]	34.86 ± 8.29 [10 to 44]	35.91 ± 7.03 [9 to 44]	<0.01 (1 vs. 2) <0.01 (1 vs. 3) <0.01 (2 vs. 3)
Tear Film Stability				
NIFBUT, seconds, mean ± SD [range]	9.37 ± 6.04 [1.10 to 17.00]	11.24 ± 5.60 [1.20 to 17.00]	10.78 ± 5.83 [1.10 to 17.00]	0.01 (1 vs. 2) <0.01 (1 vs. 3) 0.04 (2 vs. 3)
NIABUT, seconds, mean ± SD [range]	11.07 ± 4.98 [2.30 to 17.00]	12.44 ± 4.60 [2.80 to 17.00]	12.34 ± 4.66 [2.20 to 17.00]	0.11 (1 vs. 2) <0.01 (1 vs. 3) <0.01 (2 vs. 3)
TFSE, score points, mean ± SD [range]	337.78 ± 414.084 [10 to 1800]	244.47 ± 322.37 [18 to 1747]	206.02 ± 240.44 [18 to 1325]	<0.01 (1 vs. 2) <0.01 (1 vs. 3) 0.09 (2 vs. 3)
Tear Film Quantity				
CTMH, mm, mean ± SD [range]	0.44 ± 0.21 [0.09 to 1.04]	0.44 ± 0.20 [0.11 to 1.05]	0.44 ± 0.18 [0.14 to 1.02]	0.38 (1 vs. 2) 0.87 (1 vs. 3) 0.16 (2 vs. 3)
TTMH, mm, mean ± SD [range]	0.54 ± 0.28 [0.16 to 1.55]	0.51 ± 0.25 [0.09 to 1.04]	0.51 ± 0.23 [0.15 to 1.47]	0.05 (1 vs. 2) 0.21 (1 vs. 3) 0.15 (2 vs. 3)
Surface Evaluation				
OSIE Type 1, percentage, mean ± SD [range]	7.26 ± 7.86 [0 to 53]	6.11 ± 6.29 [0 to 45]	5.05 ± 4.77 [0 to 24]	0.11 (1 vs. 2) <0.01 (1 vs. 3) 0.11 (2 vs. 3)
OSIE capture time, seconds, mean ± SD [range]	130.38 ± 10.34 [107 to 150]	130.18 ± 9.97 [112 to 153]	129.15 ± 8.92 [110 to 150]	0.28 (1 vs. 2) 0.14 (1 vs. 3) 0.20 (2 vs. 3)

DED dry eye disease, EFT Eye Fitness Test, CTMH Central Tear Meniscus Height (below iris), NIABUT Non-Invasive Average Break-Up Time, NIFBUT Non-Invasive First Break-Up time, OSIE Type 1 ocular surface inflammatory evaluation (with fluorescein sodium and oxybuprocaine hydrochloride), SD standard deviation, TFSE tear film surface evaluation, TTMH Thinnest Tear Meniscus Height. ^a^ W of Wilcoxon.

## Data Availability

The raw data supporting the conclusions of this article will be made available by the authors on request.

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
