# Peer review of "Intense Pulsed Light Therapy for Dry Eye Disease: Analyzing Temporal Changes in Tear Film Stability and Ocular Surface between IPL Sessions"

_healthcare, 2024, doi:10.3390/healthcare12111119_

Round 1
Reviewer 1 Report
Comments and Suggestions for Authors
Comments and Suggestions for Authors
In the present study, the authors aim to assess the efficacy of IPL therapy on the ocular status of participants with Meibomian Gland Dysfunction (MGD) across multiple sessions. While the topic is highly interesting, the manuscript requires a thorough revision regarding the information provided. Please find below some comments that I hope will help enhance the readability and scientific impact of the manuscript.
General Comments:
There is a notable lack of references or justification for methods or procedures throughout the entire manuscript. The rationale behind each decision and design choice should be clearly stated.
The abstract needs a complete revision, as it does not accurately reflect the study; neither materials nor methods are reported nor are real results displayed here (see specific comments below).
There is significant missing information regarding diagnostic, classification, and measurement methods, which undermines the value of the manuscript's real impact. Based on the aforementioned issues, it is difficult to make a true assessment of the scientific impact of the design and results.
Specific Comments:
Line 22: I recommend employing a more scientific writing style without using "we" or "our" (also in lines 94, 105, 158, 166, 260...).
Lines 23-25: There is no need to address this issue in the abstract. As the abstract has a word limitation, I suggest utilizing these words to enhance the reported results (see comment below).
Lines 25-27: This information is unnecessary in the abstract; it belongs to the introduction, not the materials and methods section of the abstract.
Lines 29-30: The abstract should highlight the main findings based on statistically reported data (p-values, mean values, etc.) to provide the reader with a quick overview of the study without needing to refer to the main text.
Line 42: Dry Eye Disease (DED) is now recognized as a disease, not a syndrome, as generally accepted by leading worldwide ocular surface institutes (See TFOS and Asia Dry Eye Society (ADES) principles). I recommend revising the entire manuscript to reflect this according to international standards.
Line 51: Reference needed (PMID: 28939118).
Line 62: Reference needed (PMID: 37026281).
Line 103: Since the study has concluded, I recommend reporting the aim in the past tense.
Lines 103-107: The aims should be simplified and not presented in two different ways within the same paragraph.
Line 128: I suggest adding the sample size here instead of in line 198.
Line 128: How were these participants diagnosed with MGD? If the criteria were based on TFOS principles, this should be clearly stated. Additionally, there is a lack of references throughout this paragraph that should be addressed (from 128-135). Were other comorbidities or confounding factors such as contact lenses or medications considered in the inclusion criteria, alongside dermatological contraindications? This is an important aspect that should be justified by the authors (See TFOS Lifestyles reports).
Line 141-142: A reference for the categorization system is required.
Lines 154 vs. 159: Brand and manufacturer should be consistently reported throughout the manuscript.
Lines 160-161: A reference to this questionnaire/survey is required; in DED diagnosis, there are several tests recommended and validated, and from my experience, this is not included. This should be justified, as it is a significant shortcoming of the study (DED diagnosis is based on both validated signs and symptoms).
Line 169: A reference for the device is required.
Lines 169-173: This information should be either moved or briefly introduced in the study design section (lines 110-117) for improved manuscript flow.
Lines 188-196: While this seems correct, I recommend adding references to the statistical design.
Lines 207-212: There are a high number of acronyms here that have been previously introduced. Acronyms should only be introduced upon their first appearance.
Line 216: OSDI? This is the first mention of this questionnaire, and it is in the results section. Please conduct a thorough review of the materials and methods section to provide all necessary information regarding the employed materials and methods.
Author Response
Reviewer 1
#RV1_0: In the present study, the authors aim to assess the efficacy of IPL therapy on the ocular status of participants with Meibomian Gland Dysfunction (MGD) across multiple sessions. While the topic is highly interesting, the manuscript requires a thorough revision regarding the information provided. Please find below some comments that I hope will help enhance the readability and scientific impact of the manuscript.
#AU1_0: Thank you for your valuable feedback and the opportunity to enhance our manuscript. We appreciate your interest in our study on the efficacy of IPL therapy for Meibomian Gland Dysfunction (MGD) and acknowledge your concerns regarding the need for a thorough revision to improve both readability and scientific impact. We are committed to addressing each of your comments meticulously to clarify our methodology, augment our data presentation, and refine our analysis. We believe these revisions will significantly enhance the manuscript's contribution to the field and its suitability for publication.
General Comments:
#RV1_1: There is a notable lack of references or justification for methods or procedures throughout the entire manuscript. The rationale behind each decision and design choice should be clearly stated.
#AU1_1: We appreciate the reviewer's feedback on the need for more references and justification for the methods and procedures used in our study. In response, we have revised the manuscript to include additional references and detailed explanations for each decision and design choice. This includes providing scientific rationale and citing relevant studies that support our methodology. We believe these additions will enhance the clarity and rigor of our manuscript.
#RV1_2: The abstract needs a complete revision, as it does not accurately reflect the study; neither materials nor methods are reported nor are real results displayed here (see specific comments below).
#AU1_2: We appreciate the reviewer's comments regarding the abstract. We acknowledge that the abstract needs to accurately reflect the study, including a summary of the materials and methods used, as well as the key results obtained. In response, we have revised the abstract to provide a clearer and more comprehensive overview of the study, ensuring that it includes essential details about the methods and highlights the main findings. We believe this revision will provide a more accurate and informative summary of our research.
#RV1_3: There is significant missing information regarding diagnostic, classification, and measurement methods, which undermines the value of the manuscript's real impact. Based on the aforementioned issues, it is difficult to make a true assessment of the scientific impact of the design and results.
#AU1_3: We appreciate the reviewer's feedback highlighting the need for detailed information on diagnostic, classification, and measurement methods. In response, we have thoroughly revised and expanded the methods section of the manuscript. The revised section now includes comprehensive details on the diagnostic criteria for DED, the classification of patient categories based on tear film evaluations, and the specific measurement techniques employed using the Tearcheck® device and SCHWIND SIRIUS corneal pachymetry and topography device. We believe these enhancements significantly improve the clarity and scientific rigor of our study, thereby allowing for a more accurate assessment of the design and results.
Specific Comments:
#RV1_4: Line 22: I recommend employing a more scientific writing style without using "we" or "our" (also in lines 94, 105, 158, 166, 260...).
#AU1_4: Thank you for your suggestion. The manuscript has been revised to employ a more scientific writing style, avoiding the use of "we" or "our" in the specified lines and throughout the text.
#RV1_5: Lines 23-25: There is no need to address this issue in the abstract. As the abstract has a word limitation, I suggest utilizing these words to enhance the reported results (see comment below).
#AU1_5: Thank you for the feedback. The issue has been removed from the abstract. The words have been reallocated to enhance the reported results, as suggested.
#RV1_6: Lines 25-27: This information is unnecessary in the abstract; it belongs to the introduction, not the materials and methods section of the abstract.
#AU1_6: Thank you for the suggestion. The information has been relocated to the introduction, and the abstract has been revised accordingly. This issue is now resolved with the new abstract rewrite.
#RV1_7: Lines 29-30: The abstract should highlight the main findings based on statistically reported data (p-values, mean values, etc.) to provide the reader with a quick overview of the study without needing to refer to the main text.
#AU1_7: Thank you for the feedback. The abstract has been revised to highlight the main findings with statistically reported data, including p-values and mean values, to provide a concise overview of the study.
#RV1_8: Line 42: Dry Eye Disease (DED) is now recognized as a disease, not a syndrome, as generally accepted by leading worldwide ocular surface institutes (See TFOS and Asia Dry Eye Society (ADES) principles). I recommend revising the entire manuscript to reflect this according to international standards.
#AU1_8: Thank you for pointing this out. The manuscript has been revised to reflect that Dry Eye Disease (DED) is recognized as a disease, in accordance with the principles of leading worldwide ocular surface institutes such as TFOS and the Asia Dry Eye Society (ADES).
#RV1_9: Line 51: Reference needed (PMID: 28939118).
#AU1_9: Thank you for your suggestion. The reference has been added at the specified line (PMID: 28939118).
#RV1_10: Line 62: Reference needed (PMID: 37026281).
#AU1_10: Thank you for your suggestion. The reference has been added at the specified line (PMID: 37026281).
#RV1_11: Line 103: Since the study has concluded, I recommend reporting the aim in the past tense.
#AU1_11: Thank you for the recommendation. The aim of the study has been revised to be reported in the past tense, reflecting that the study has concluded.
#RV1_12: Lines 103-107: The aims should be simplified and not presented in two different ways within the same paragraph.
#AU1_12: Thank you for the suggestion. The aims have been simplified to avoid presenting them in two different ways within the same paragraph. The revised purpose is as follows:
“The purpose of this study was to systematically investigate the temporal changes in tear film stability and ocular surface health in patients with DED undergoing IPL therapy over multiple treatment sessions, with the aim of elucidating the efficacy of IPL therapy in improving the symptoms and underlying causes of DED between treatment intervals.”
#RV1_13: Line 128: I suggest adding the sample size here instead of in line 198.
#AU1_13: Thank you for the suggestion. The sample size has been added at line 128, as recommended, instead of at line 198.
#RV1_14: Line 128: How were these participants diagnosed with MGD? If the criteria were based on TFOS principles, this should be clearly stated. Additionally, there is a lack of references throughout this paragraph that should be addressed (from 128-135). Were other comorbidities or confounding factors such as contact lenses or medications considered in the inclusion criteria, alongside dermatological contraindications? This is an important aspect that should be justified by the authors (See TFOS Lifestyles reports).
#AU1_14: Thank you for your thorough review and valuable suggestions. The following revisions have been made to address your concerns:
- The diagnosis of MGD in participants has been clarified to state that it was based on TFOS principles.
- Additional references have been added throughout the paragraph (lines 128-135) to support the methodology.
- The exclusion criteria have been expanded to consider other comorbidities or confounding factors such as contact lenses, medications, and dermatological contraindications, following the TFOS Lifestyle reports [1–5].
These adjustments aim to provide a comprehensive understanding of the participant selection process and ensure the study's rigor and alignment with established standards.
#RV1_15: Line 141-142: A reference for the categorization system is required.
#AU1_15: Thank you for the suggestion. A reference for the categorization system has been added to support the methodology [6].
#RV1_16: Lines 154 vs. 159: Brand and manufacturer should be consistently reported throughout the manuscript.
#AU1_16: Thank you for pointing this out. The brand and manufacturer details have been made consistent throughout the manuscript.
#RV1_17: Lines 160-161: A reference to this questionnaire/survey is required; in DED diagnosis, there are several tests recommended and validated, and from my experience, this is not included. This should be justified, as it is a significant shortcoming of the study (DED diagnosis is based on both validated signs and symptoms).
#AU1_17: Thank you for your feedback. The Eye Fitness Test (EFT) was selected as it is the default assessment tool integrated within the Tearcheck® device used in this study. This choice was driven by the need for consistency in data collection across all participants. The design of the EFT closely mirrors that of the OSDI questionnaire, with questions that are essentially the same. The primary difference lies in the inverse scoring system. Consequently, a higher score in the EFT indicates better ocular health, whereas in the OSDI, a higher score indicates worse ocular health. This inverse scoring does not affect the comparability of the results, as the variation in outcomes is consistent due to the identical nature of the questions asked.
We recognize the importance and widespread acceptance of validated symptomatology tests like OSDI and DEQ-5 in the scientific community. We acknowledge that using the EFT is a limitation, and this has been addressed in the limitations section of our manuscript, discussing its implications for interpreting the study results.
“One limitation of this study is the use of the Eye Fitness Test (EFT) as the primary assessment tool for symptom severity, instead of more widely validated tests such as the OSDI or DEQ-5. The EFT was selected because it is the default assessment tool integrated within the Tearcheck® device, ensuring consistency in data collection. While the questions in the EFT mirror those in the OSDI, the scoring system is inverse. Despite this, the comparability of the results remains unaffected due to the identical nature of the questions. We acknowledge that this choice may affect the generalizability of our findings, and future studies should consider using widely validated symptomatology tests to align with common clinical practices and enhance comparability across studies.”
#RV1_18: Line 169: A reference for the device is required.
#AU1_18: Thank you for the suggestion. The reference for the device has been added [7].
#RV1_19: Lines 169-173: This information should be either moved or briefly introduced in the study design section (lines 110-117) for improved manuscript flow.
#AU1_19: Thank you for the suggestion. The information regarding the SCHWIND SIRIUS device has been moved to the study design section (lines 110-117) to improve the manuscript flow.
#RV1_20: Lines 188-196: While this seems correct, I recommend adding references to the statistical design.
#AU1_20: Thank you for your suggestion. A statistical guide reference has been added to support the statistical design mentioned in lines 188-196 [8].
#RV1_21: Lines 207-212: There are a high number of acronyms here that have been previously introduced. Acronyms should only be introduced upon their first appearance.
#AU1_21: Thank you for your feedback. The manuscript has been revised to ensure that acronyms are only introduced upon their first appearance, reducing redundancy and improving readability.
#RV1_22: Line 216: OSDI? This is the first mention of this questionnaire, and it is in the results section. Please conduct a thorough review of the materials and methods section to provide all necessary information regarding the employed materials and methods.
#AU1_22: Thank you for pointing this out. The mention of OSDI in line 216 was a typo and should have been EFT.
References
- Gomes, J.A.P.; Azar, D.T.; Baudouin, C.; Bitton, E.; Chen, W.; Hafezi, F.; Hamrah, P.; Hogg, R.E.; Horwath-Winter, J.; Kontadakis, G.A.; et al. TFOS Lifestyle: Impact of Elective Medications and Procedures on the Ocular Surface. Ocul. Surf. 2023, 29, 331–385, doi:10.1016/j.jtos.2023.04.011.
- Jones, L.; Efron, N.; Bandamwar, K.; Barnett, M.; Jacobs, D.S.; Jalbert, I.; Pult, H.; Rhee, M.K.; Sheardown, H.; Shovlin, J.P.; et al. TFOS Lifestyle: Impact of Contact Lenses on the Ocular Surface. Ocul. Surf. 2023, 29, 175–219, doi:10.1016/j.jtos.2023.04.010.
- Downie, L.E.; Britten-Jones, A.C.; Hogg, R.E.; Jalbert, I.; Li, T.; Lingham, G.; Liu, S.-H.; Qureshi, R.; Saldanha, I.J.; Singh, S.; et al. TFOS Lifestyle - Evidence Quality Report: Advancing the Evaluation and Synthesis of Research Evidence. Ocul. Surf. 2023, 28, 200–212, doi:10.1016/j.jtos.2023.04.009.
- Craig, J.P.; Alves, M.; Wolffsohn, J.S.; Downie, L.E.; Efron, N.; Galor, A.; Gomes, J.A.P.; Jones, L.; Markoulli, M.; Stapleton, F.; et al. TFOS Lifestyle Report Introduction: A Lifestyle Epidemic - Ocular Surface Disease. Ocul. Surf. 2023, 28, 304–309, doi:10.1016/j.jtos.2023.04.014.
- Craig, J.P.; Alves, M.; Wolffsohn, J.S.; Downie, L.E.; Efron, N.; Galor, A.; Gomes, J.A.P.; Jones, L.; Markoulli, M.; Stapleton, F.; et al. TFOS Lifestyle Report Executive Summary: A Lifestyle Epidemic - Ocular Surface Disease. Ocul. Surf. 2023, 30, 240–253, doi:10.1016/j.jtos.2023.08.009.
- Jimbow, K.; Quevedo, W.C.J.; Fitzpatrick, T.B.; Szabo, G. Some Aspects of Melanin Biology: 1950-1975. J. Invest. Dermatol. 1976, 67, 72–89, doi:10.1111/1523-1747.ep12512500.
- Mansoori, T.; Balakrishna, N. Repeatability and Agreement of Central Corneal Thickness Measurement with Non-Contact Methods: A Comparative Study. Int. Ophthalmol. 2018, 38, 959–966, doi:10.1007/s10792-017-0543-1.
- Chatfield, C. A Guide to Statistics. Physiotherapy 1982, 68, 227–230.
Reviewer 2 Report
Comments and Suggestions for Authors
Line 128: what do you mean by symptomatic MGD, please define and what considered for symptomatic MGD?
Line 138 : Add the image of IPL machine which is used in the study
Line 141 : what is the wavelength and fluence used while providing the IPL treatment?
Line 153 : After each IPL treatment session which safety measure applied for the participants, explain in detail
Line 160 : EFT questionnaire should be attached with manuscript for understanding the nature of the questions.
Line 163 : CTMH and TTMH measurement were taken from which instrument? and how the procedure performed?
Line 164: how the measurement performed for TFSE, and OSIE not discussed and what is the acceptable values?
Line 167 : Explain the procedure for NIFBUT and NIABUT and acceptable measurement needs to discuss in detail ?
Line 169: Why the Meibomian gland secretion and expressibilty grading were included as a clinical variables, without this how do you say MGD decreased with IPL treatment
Previous studies shown Meibomian gland expression (MGX) treatment following IPL therapy much acceptable and improved the DED and MGD , why MGX was not performed.
Author Response
Reviewer 2
#RV2_1: Line 128: what do you mean by symptomatic MGD, please define and what considered for symptomatic MGD?
#AU2_1: We appreciate the reviewer's request for clarification regarding the term "symptomatic MGD." In our study, "symptomatic MGD" refers to patients who were clinically diagnosed with MGD and who also reported symptoms associated with DED, such as burning, dryness, and similar discomforts. These symptoms were considered in conjunction with clinical signs to identify patients with symptomatic MGD.
“The cohort comprised adult individuals diagnosed with symptomatic MGD who underwent IPL therapy at the clinic during the specified study period. "Symptomatic MGD" refers to patients who were detected with Meibomian Gland Dysfunction (MGD) and also reported symptoms associated with Dry Eye Disease (DED), including burning, dryness, and similar discomforts.”
#RV2_2: Line 138 : Add the image of IPL machine which is used in the study
#AU2_2: We appreciate the reviewer's suggestion to include an image of the IPL machine used in our study. We have now added a photograph of the IPL machine to Figure 2 to provide a clearer understanding of the equipment utilized in our research.
#RV2_3: Line 141 : what is the wavelength and fluence used while providing the IPL treatment?
#AU2_3: We appreciate the reviewer's query regarding the specific parameters of the IPL treatment used in our study.
“The IPL treatment was administered using a wavelength range of 500-1200 nm and a fluence of 13 J/cm².”
#RV2_4: Line 153 : After each IPL treatment session which safety measure applied for the participants, explain in detail
#AU2_4: We appreciate the reviewer's interest in the safety measures applied during our study. After each IPL treatment session, the following safety measures were applied for all participants:
During treatment, skin lesions and moles have been covered with special patches.
Protective Eyewear: Participants were instructed to wear sunglasses to protect their eyes from bright light and UV exposure.
Sun Exposure: Participants were advised to avoid exposing their skin to strong sunlight, sunbeds, or self-tan for at least 2 weeks after treatment.
Heat Exposure: Participants were instructed to avoid excess heat, such as long baths, spas, steam rooms, and saunas, for at least 24 hours or longer if the skin was still red or recovering.
Chlorine Exposure: Participants were advised to avoid activities involving chlorine, such as swimming, for 48 hours post-treatment.
Post-Treatment Instructions: Detailed post-treatment care instructions were provided to ensure proper skin care and protection.
We include in the manuscript:
“After each IPL treatment session, several safety measures were applied to ensure participant safety and comfort. Participants were instructed to wear sunglasses to protect their eyes from bright light and UV exposure. They were advised to avoid exposing their skin to strong sunlight, sunbeds, or self-tan for at least 2 weeks after treatment. Participants were also instructed to avoid excess heat, such as long baths, spas, steam rooms, and saunas, for at least 24 hours or longer if the skin was still red or recovering. Additionally, they were advised to avoid activities involving chlorine, such as swimming, for 48 hours post-treatment. Detailed post-treatment care instructions were provided to ensure proper skin care and protection.”
#RV2_5: Line 160 : EFT questionnaire should be attached with manuscript for understanding the nature of the questions.
#AU2_5: We appreciate the reviewer's suggestion to include the Eye Fitness Test (EFT) questionnaire in the manuscript. We have now attached the EFT questionnaire as an appendix to provide a comprehensive understanding of the nature of the questions used in our study.
#RV2_6: Line 163 : CTMH and TTMH measurement were taken from which instrument? and how the procedure performed?
#AU2_6: We appreciate the reviewer's inquiry regarding the measurement of Central Tear Meniscus Height (CTMH) and Thinnest Tear Meniscus Height (TTMH). These measurements were taken using the TearCheck® device. The Tearcheck system measures tear meniscus height (TMH) by first instilling fluorescein dye into the eye to highlight the tear film. It then captures high-resolution images and uses automated algorithms to detect and measure the height of the tear meniscus from these images. This provides a precise and efficient assessment of TMH.
“Central Tear Meniscus Height (CTMH) and Thinnest Tear Meniscus Height (TTMH) measurements were taken using the TearCheck® device. The procedure involved measuring the tear meniscus height by first instilling fluorescein dye into the eye to highlight the tear film. It then captures high-resolution images and uses automated algorithms to detect and measure the height of the tear meniscus from these images.”
#RV2_7: Line 164: how the measurement performed for TFSE, and OSIE not discussed and what is the acceptable values?
#AU2_7: We appreciate the reviewer's inquiry regarding the measurement of Tear Film Surface Evaluation (TFSE) and Ocular Surface Inflammatory Evaluation (OSIE). These measurements were performed using standardized procedures to ensure accurate assessment of tear film stability and ocular surface inflammation.
The TFSE allows for the assessment of micro-deformations on the tear film surface, reflecting its instability. During a 10-second imaging period, the tear film of a healthy eye shows very few, low-intensity movements. In contrast, a patient with dry eye syndrome, linked to a lipid tear film deficiency, exhibits higher micro-deformations. Patients are categorized into four groups based on the number and intensity of these micro-deformations.
The OSIE involves applying fluorescein dye to the ocular surface, which adheres to areas with surface alterations due to inflammation. The evaluation occurs 120 seconds post-instillation, allowing for the natural elimination of the dye through the tear ducts. In healthy eyes, fluorescein disappears, while in dry eye syndrome, it remains in damaged areas, indicating inflammation.
Acceptable values for TFSE and OSIE depend on the severity and progression of the dry eye condition, with specific criteria outlined below.
“Tear Film Surface Evaluation (TFSE):
TFSE assesses the micro-deformations on the tear film surface, which reflect tear film instability. These deformations are presented in terms of number and intensity during a 10-second imaging period. The tear film of a healthy eye shows very few, low-intensity movements, whereas a patient with dry eye syndrome, linked to a deficiency in the lipid tear film component, shows higher micro-deformations. The frequency and intensity of these deformations are observed throughout the imaging period, allowing for the classification of patients into four categories:
Category 1: Healthy patient with very few, low-intensity micro-deformations.
Category 2: Significant number of micro-deformations grouped towards the end of the 10-second acquisition, regardless of intensity.
Category 3: Early onset of micro-deformations with minimal evolution over the 10-second period.
Category 4: Early onset of micro-deformations with increasing number and intensity over the 10-second period.
The higher the category, the greater the lipid deficiency, with Category 4 patients experiencing the most significant discomfort and unfavorable progression without treatment. Compared to Non-Invasive Break-Up Time (NIBUT), TFSE provides a more detailed evolution of tear film behavior over time, showing finer nuances of the tear film surface. The device converts the grade into a score ranging from 18 to 1200 points, providing a detailed and quantifiable assessment of the tear film.
Ocular Surface Inflammatory Evaluation (OSIE):
OSIE utilizes fluorescein dye, which adheres to areas of the ocular surface with alterations due to inflammation. The evaluation is conducted 120 seconds after instilling fluorescein, allowing for its natural elimination through the tear ducts. In a healthy patient, fluorescein disappears from the ocular surface, showing 0% residual fluorescence. In contrast, in patients with dry eye syndrome, the dye remains in the affected areas beyond 120 seconds, indicating inflammation. The accuracy of this examination relies on the practitioner's selections and the use of adjustment sliders to evaluate these inflammatory zones accurately.”
#RV2_8: Line 167 : Explain the procedure for NIFBUT and NIABUT and acceptable measurement needs to discuss in detail ?
#AU2_8: We appreciate the reviewer's request for a detailed explanation of the procedures for measuring Non-Invasive First Break Up Time (NIFBUT) and Non-Invasive Average Break Up Time (NIABUT). These measurements were conducted using the SCHWIND SIRIUS Corneal pachymetry and topography device (SCHWIND eye-tech-solutions GmbH, Kleinostheim, Germany). The NIFBUT and NIABUT provide critical assessments of tear film stability, which are essential in evaluating the ocular surface condition in patients with dry eye disease.
“NIFBUT:
- NIFBUT is the time interval between the last complete blink and the first appearance of a dry spot or discontinuity in the tear film.
- The patient is asked to blink naturally, then to keep their eyes open for as long as possible while the device records the tear film.
- The SCHWIND SIRIUS device projects a series of concentric rings onto the cornea and captures high-resolution images to detect the first break in the tear film. The time at which the first break occurs is recorded as the NIFBUT.
NIABUT:
- NIABUT measures the average time taken for multiple tear film break-ups to occur across the corneal surface.
- Following the same initial procedure, the device continuously monitors the tear film over a specified period, capturing the times at which multiple breaks in the tear film appear.
- The average time of these break-ups is calculated and recorded as the NIABUT.
Acceptable Measurement Values:
For healthy individuals, NIFBUT values typically range above 10 seconds, indicating a stable tear film. NIABUT values generally above 15 seconds in healthy eyes. Lower values for NIFBUT and NIABUT indicate reduced tear film stability, which is often observed in patients with dry eye disease. The device has a cut-off superior point of 17 seconds.”
#RV2_9: Line 169: Why the Meibomian gland secretion and expressibilty grading were included as a clinical variables, without this how do you say MGD decreased with IPL treatment
#AU2_9: We appreciate the reviewer's concern regarding the inclusion of Meibomian gland secretion and expressibility grading as clinical variables. In our study, the primary focus was on tear film variables, such as tear film stability and break-up time, to indirectly assess the improvement in Meibomian Gland Dysfunction (MGD) following IPL treatment. We acknowledge that a direct analysis of Meibomian gland secretion and expressibility was not performed. This is recognized as a limitation of our study, and we plan to include these variables in future research to provide a more comprehensive evaluation of MGD improvement with IPL treatment.
“Our study primarily focused on tear film variables, such as tear film stability and break-up time, to indirectly assess the improvement in MGD following IPL treatment. While increased tear film stability and longer break-up times suggest an improvement in MGD, we did not directly analyze Meibomian gland secretion and expressibility. This omission is a limitation of our study.”
#RV2_10: Previous studies shown Meibomian gland expression (MGX) treatment following IPL therapy much acceptable and improved the DED and MGD , why MGX was not performed.
#AU2_10: We appreciate the reviewer's comment regarding the inclusion of Meibomian gland expression (MGX) following IPL therapy. While we agree that previous studies have shown the benefits of combined IPL and MGX treatment in improving DED and MGD, our study aimed to evaluate the effect of IPL alone. By excluding MGX, we sought to eliminate any confounding factors and avoid potential bias, thereby isolating the specific impact of IPL therapy on tear film stability and dry eye symptoms.
Reviewer 3 Report
Comments and Suggestions for Authors
Is it possible to show a photo of the patient just prior to treatment with protective eyewear? Can the authors provide their opinion regarding IPL effects on inflammation in the tear film itself; is measurement of markers or assessment of the presence of white blood cells or cellular debris possible in a future paper?
Author Response
Reviewer 3
#RV3_1: Is it possible to show a photo of the patient just prior to treatment with protective eyewear?
#AU3_1: We have included a series of photographs in Figure 1 to provide a comprehensive visual representation of the patient's condition and treatment process as requested.
Specifically, Figure 1 now includes:
Figure 1A: A photograph of the patient just prior to treatment, wearing protective eyewear shields.
Figure 1B: A photograph of the patient during the application of IPL treatment.
Figure 1C: A photograph of the patient immediately after the removal of the conductive gel.
These additions to Figure 1 aim to enhance the clarity of the treatment procedure and the patient's condition at each stage. We hope these visual aids address the reviewer's request effectively.
#RV3_2: Can the authors provide their opinion regarding IPL effects on inflammation in the tear film itself; is measurement of markers or assessment of the presence of white blood cells or cellular debris possible in a future paper?
#AU3_2: We appreciate the reviewer's insightful question regarding the effects of IPL on inflammation in the tear film. In our study, we focused on assessing inflammation through the staining process of the cornea. Specifically, in the context of Dry Eye Disease (DED), the Ocular Surface Inflammatory Evaluation (OSIE) provides an essential measure of disease activity by detecting and quantifying ocular surface staining. This procedure utilizes fluorescein dye, a vital diagnostic tool that adheres to altered epithelial areas, indicating potential inflammation. Through a detailed analysis of fluorescein staining patterns, OSIE enables clinicians to visualize and assess the severity of ocular surface damage. This approach enhances our understanding of the ocular surface condition in DED and aids in tailoring individualized treatment strategies aimed at promoting ocular surface healing.
However, we did not measure white blood cells, cellular debris, osmolarity, or metalloprotease levels in our study. We acknowledge this as a limitation and plan to address these aspects in future research.